# Data Intelligence in Public Transportation: Sustainable and Equitable Solutions to Urban Modals in Strategic Digital City Subproject

Luis André Wernecke Fumagalli [1,2] , Denis Alcides Rezende [1,*] and Thiago André Guimarães [3,4]

1   Master's and Doctoral Program in Urban Management, Strategic Digital City Research Group (CNPq), Pontifícia Universidade Católica do Paraná (PUCPR), Curitiba 80215-901, Brazil; luis.fumagalli@fae.edu
2   MBA Department, FAE Business School (FAE), Curitiba 80010-100, Brazil
3   Business School, Federal University of Paraná (UFPR), Curitiba 80210-170, Brazil; thiagoandre@ufpr.br
4   Technical and Technological Education Department, Federal Institute of Science and Technology of Paraná (IFPR), Curitiba 80230-150, Brazil
*   Correspondence: denis.rezende@pucpr.br

**Abstract:** Transport infrastructure investments must be linked to the public transport demand strategically. User behavior and decision-making process bring several possible alternative transportation options due to a series of factors that define it. Municipalities must manage these factors to promote equal and sustainable transport solutions through urban infrastructure, public transport competitiveness, and attractiveness, and fossil fuels use and pricing policy. The research objective is to determine these factors to monitor and use them for citizens' benefit by means of analytical tools and methods to gain a superior knowledge of reality with a focus on improving investments and services in an agile and efficient manner. Methodologically, the number of passengers of main Curitiba's (Brazil) bus rapid transit (BRT) lines is operated in two linear regression models combined with the number of private vehicles, public transport fare, and fuel price for the period between January/2010 and December/2019. Research analysis indicates direct causal relationships between the studied factors and that the necessary data for decision-making is available in the government information systems. In conclusion, urban management and strategic digital city project can be more balanced and assertive in transport infrastructure investments and citizen services provision.

**Keywords:** public transport; public administration; public investments; sustainable urban development; strategic digital city

## 1. Introduction

Many cities are growing faster than their capacity to manage urban form, structure, and public services. Security, education, healthcare system, and transportation are the main issues to municipal administration, especially in big urban centers where access to those essential services is unequally higher and expensive. Smart city, strategic digital city projects, and public transport are strategic issues because they are mainly responsible for all other public services integration. And crucial for economic development because workers and students are transport dependent to move safely and comfortably to perform their daily activities. The systems perspective allied to new institutional requirements should conduce public transport to affordable and sustainable solutions. The new mobility, city logistics, livability, and intelligent system management are fundamental elements to achieve it successfully [1]. Since all needed information is available almost in real-time, municipalities shall define their urban transport policies to avoid decongestion, energy waste, emissions reduction, and saving money to invest in other critical areas and combating pollution [2,3].

Space and budget are two fundamental limitations for municipal transport strategic management due to buses, cars, and motorcycles sharing the same routes simultaneously.

The city does not have sufficient resources to invest in all the infrastructure that each vehicle requires. The exclusive use of public transport causes overcrowding, especially during rush hours and does not cover all transportation needs in many city areas where the density of lines is lower and distances to access them are too long to be covered by foot or bicycle. That makes many passengers prefer to buy or use private motorized vehicles.

However, increasing the number of vehicles to replace public transport causes traffic jams, raises air and noise pollution and causes accidents with significant physical and material damage. Bicycles, in turn, are recognized as a clean, cheap, and sustainable transport mode. However, they still depend on dedicated road infrastructure since they cannot simply move in between other vehicles without cyclists incurring risks. Proper integration avoids that motor vehicle traffic does not suffer a drastic reduction in the number of available lanes and speed, causing inefficiencies and dissatisfaction for users.

Large cities have lost public transport users due to unreliability, lack of comfort, and the fare price charged. The effects of different urban transport modes are interdependent and systemic, and there is no alternative to choosing just one or the other without causing imbalances and negative consequences in terms of operational and economic efficiency and social equitability [1]. Cities need to invest in a planned and strategic direction to establish a clear transport policy to integrate all modes and other innovative solutions. Municipalities must work to turn public transport competitive and attractive, avoiding public roads stopping by the excess of private vehicles and bicycles to have their own space, safe and connected with public transport [4,5]. The inherent conflict between wealth generation and high efficiency in resources utilization with sustainable solutions must be dealt with in the political sphere and administrated by the city authorities [6].

Based on this scenario, the present article's objectives are to determine which factors citizens are taking to decide the transport mode among different options and conditions. Once these factors are known, it would be possible for the municipality to monitor and connect this information with administrative decisions to manage public transportation, ticket pricing, and city transport infrastructure investments. To reach this objective, two linear regression models tested the number of passengers from the six express bus rapid transit (BRT) lines with the number of registered cars, registered motorcycles, and public transport fare and fuel price. These BRT lines cross and connect the city in the most important directions and represent almost 30% of the total passengers transported every day. The observation period comprises registered data from January/2010 to December/2019.

Research results found a significant relationship between studied factors that can be controlled or at least monitored by the city administration, that are available instantly. Managing this information should result in intelligent decision-making able to define and integrate, equitably and sustainably, public transport tariffs, fuel prices, investments in road infrastructure, tax incentives to acquire alternative private vehicles or bicycles, and public transport improvements.

## 2. Literature Review

Public transport represents a critical point in decision-making by municipal governments and administrators due to local development promotion and involves continuous investments that must come from insufficient budgets [7,8]. There is pressure, on the one hand, from users who rely solely on public transport to go from home to school or work, and on the other, the claim for sustainable solutions in the social, economic, political, ecological, environmental, and local territorial spheres [9]. Regarding urban mobility and sustainability criteria [10], the ideal city would be one in which people live at a walking distance or, at most, by bicycle to their daily points of interest to work, study, shopping, groceries, access public services, and to participate in leisure and entertainment activities [11,12].

In parallel, the social status pressure to substitute public transportation with the owned vehicle, preferably the car, still exists [13]. Having the owned car or, when it is not affordable, a motorcycle, represents an icon of personal success [14] or bicycles, which can

offer higher economy, freedom, and comfort for those who use it when the geographical and climatic conditions are favorable [4].

All modes require planned, coordinated, substantial, and long-term investments in the city infrastructure to support, in a balanced and efficient way, the increase in the fleet of vehicles of all types, which, most of the time, grows at the expense of public transport use. Curitiba (PR), a city with just over 1.9 million inhabitants [15], is located in the south of Brazil, and known for its sustainable urban mobility solutions and projects [16], is facing a drop of almost 40% in the total number of passengers between the years 2010 and 2019 and its downward trend should also continue due to the pandemic effects. The result of this change in the behavior of public transport users reflects in constant congestion, even outside rush hours, and in the high number of accidents involving motorcycles and, especially, bicycles that do not have exclusive lanes for circulation [17], as with most bus lines throughout the city.

The option for bicycles is notably sustainable, non-polluting, with fewer space requirements, but demands investments in infrastructure for circulation that must integrate with public transport. Otherwise, it will exclude even more passengers from the transport system, returning to the initial problem: affecting the tariff [18]. In this way, public transport, urban mobility, and public investments in these sectors relate to municipal themes that need a holistic transport project that includes municipal strategies, municipal information systems, public services, and information technology resources application [19,20].

Many studies are available in the world literature regarding different factors that affect transport demand. The authors of [21] proposed a guide on the variables for land transport, including the effects of the fare value, of the quality of service, including time factors, security, and reliability, competition between modes, income, and car ownership, land-use policy, new transport modes, and other transport policies. The author of [22] focused on how transit price elasticities can affect passengers' decisions. The authors of [23] studied the influences of fares, level of service, salaries, and car property on transportation decisions. The author of [24] also worked on the fare and fuel prices models on transport demand.

The authors of [25] also researched the influence of economic cycles on transport demand, including the factors above-mentioned, like fare and fuel prices and private vehicles per capita, with local per capita income and unemployment rates. The authors of [26] developed a forecast model to anticipate possible moves from public transportation to another private modal improving users' behavior understanding. Remuneration, stage of life (over 49 years old), changes in mode availability, service, and travel time are relevant variables to predict migration to other modes. The authors of [27] used a dynamic panel model, and their results show that mobility behaviors are dependent on various variables that include service quality, modal price, and active population.

The author of [28] recommended that demand models include car property, fuel and tariff price, salary, and some measure of service among the explanatory variables. Lately, the author of [29] tested a model where income directly affects public transport demand and, in opposition, car ownership. In conclusion, although the findings of several previous studies suggest that demand for public transport might be falling with increased income, there is no evidence of such effects even considering 100% influence of changes in income (including changes in car ownership).

Socio-technological artifacts build and organize cities; they are fashioned, modified, and appropriated as part of a long process. Various spatial, social, economic, technological, and political contexts embed these artifacts [30]. Public transport planning is, thus, not an optimal equation result. Political considerations constrain even well-thought-out plans, and hence community structure and power relations must be considered, as well as 'softer' psychological factors [31]. The author of [32] has pointed out that sustainability and public investments are obstacles for more private-sector investments in infrastructure and stated that the transport system's operations would be the main transport trends and policies into the 21st century.

In urban centers, changes from motorized vehicles to clean transport modes, combined with land use and proper infrastructure for walking and cycling, will demand integrated planning and managing from the city hall. The result may increase physical activity and reduce non-communicable diseases (NCDs), accidents, pollution, and carbon emissions. Therefore, transport planning and urban configurations must reinforce clean and non-motorized transport alternatives [33].

The authors of [34] found a power function not limited to the political scenario about how transport demand patterns may affect the public transport offerings. Instead, they can affirm that transport users have other preferable options to quit from the public system, as per private cars and other private services that offer attractive alternatives like car-sharing [5] and transport apps. Urban management processes must include softer elements, such as consumer behavior and choice decisions, power, and conflict. Nevertheless, political influences and biases are embedded in urban transport planning that can benefit groups and jeopardize others [31].

Curitiba's transport system is dependent on the policy directions developed in the 1970s that remain working all these years. However, interaction, flexibility, and integration with other modern transport solutions and city services are urged [35]. Curitiba administrators learn and innovate incrementally through decades from their local expertise and technical capacity. City institutions responsible for the plan and supervising the city transport system worked together with the urban planning. The continuity in political administration played a decisive role in it together with land-use mobility policies [36].

Smart city and conventional digital city concepts are different from the strategic digital city concept defined by Rezende [37]. It is the application of information technology in the city's management and the information and services provided to citizens, based on the city management strategies. There are four subprojects: city strategies, cities information, public services, and information technology resources application [19,20]. It can also be considered a city public transport policy [37,38].

## 3. Research Methodology

The proposed models address quantifying the substitution effect between automobiles and motorcycle use, to the detriment of the use of the public transport system. The analysis was sought to measure the number of vehicles' (cars and motorcycles) direct impact on the reduction in demand for public transport observed throughout the historical period studied. The regression model aims to measure the determinants of passenger demand for the aggregate of the lines. Formally, there are:

T : monthly periods between January 2010 and December 2019, where $|T| = 120$.

To obtain greater representativeness of the use of the system, six-line passenger records were grouped, defining the set $\Lambda$, as follow:

$\Lambda$ : set of aggregate lines for analysis, where $|\Lambda| = 6$.

The dependent variable $y_i^t$ with $i \in \Lambda$, $t \in T$, represents the number of passengers on the line $i$ at period $t$. This representation produces a matrix of records with 720 elements, described in Table 1.

**Table 1.** Analyzed periods.

| Line | $t = 1$ (January/2010) | $t = 2$ (February/2010) | ... | $t = 120$ (December/2019) |
|---|---|---|---|---|
| $i = 1$ (203) | $y_1^1$ | $y_1^2$ | | $y_1^{120}$ |
| $i = 2$ (303) | $y_2^1$ | $y_2^2$ | | $y_2^{120}$ |
| $i = 3$ (502) | $y_3^1$ | $y_3^2$ | | $y_3^{120}$ |
| $i = 4$ (503) | $y_4^1$ | $y_4^2$ | | $y_4^{120}$ |
| $i = 5$ (602) | $y_5^1$ | $y_5^2$ | | $y_5^{120}$ |
| $i = 6$ (603) | $y_6^1$ | $y_6^2$ | | $y_6^{120}$ |

Finally, the dependent variable aggregate for each analyzed period is obtained by directly adding the observed values in each line, as shown as follow:

$$Y^t = \sum_{i \in \Lambda} y_i^t = y_1^t + y_2^t + y_3^t + y_4^t + y_5^t + y_6^t \tag{1}$$

In order to explain passengers' demand, two independent variables were tabulated, as detailed below:

$X_{car}^t$ : number of cars registered in Curitiba in period $t$. Due to the substitution relation with the use of public transport, the hypothesis is that the impact is negative.

$X_{mot}^t$ : number of motorcycles registered in Curitiba in period $t$. Due to the substitution relation with the use of public transport, the hypothesis is that the impact is negative.

Based on demand dynamics, data were operated in two modal replacement models. This analysis objective is to verify if there is a fact of the mode of transport substitution and, additionally, to detect which one is capturing systems' passengers. A simple linear regression model was operated separately for cars and motorcycles to investigate which modality affects the system demand.

### 3.1. Relationship between Number of Registered Cars and System Demand

Analysis model:

$$M1: Y^t = \beta_0 + \beta_1 X_{car}^t \tag{2}$$

$n \geq 20$ is required for each predictor variable. A single predictor variable and 120 ($|T| = 120$) observations satisfy this requirement, consequently. The linear relationship between the independent ($X_{car}^t$) and dependent ($Y^t$) variables is observable in Figure 1.

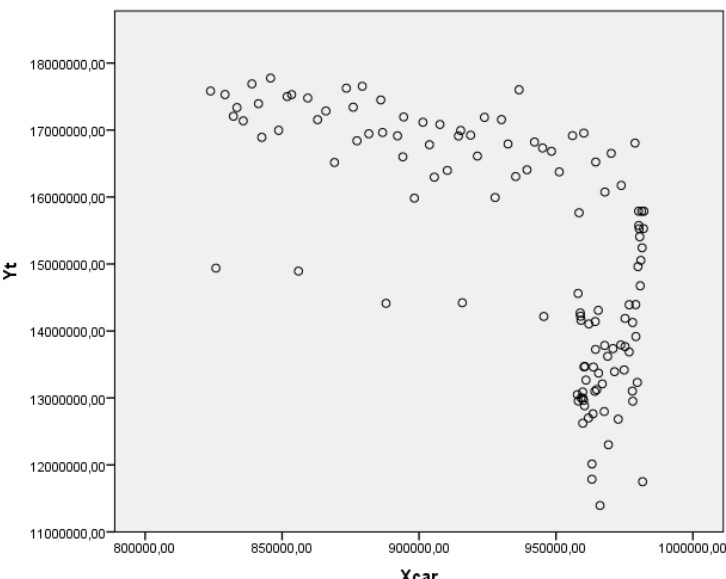

**Figure 1.** Linear relationship between independent variable ($X_{car}^t$) and dependent variable ($Y^t$).

It is possible to observe that the relationship is not perfectly linear between the variables, although there is a visual relationship. Pearson's coefficient is moderate to strong ($-0.671$) and in fact negative [39]. That means, as the number of registered cars increases, the number of passengers decreases. The correlation e is statistically significant ($p = 0.000 < 0.005$) (Table 2).

**Table 2.** Correlation analysis results.

| | | *Yt* | *Xcar* |
|---|---|---|---|
| Pearson Correlation | *Yt* | 1.000 | **−0.671** |
| | *Xcar* | **−0.671** | 1.000 |
| Sig. (1-tailed) | *Yt* | | 0.000 |
| | *Xcar* | 0.000 | |
| N | *Yt* | 120 | 120 |
| | *Xcar* | 120 | 120 |

Regarding independent residues, a model's prerequisites do not have autocorrelation between the residuals (difference between the observed value and the value predicted by the model). In this case, the Durbin–Watson test must have a value between 1.0 and 2.5. The data show that the R-squared is 0.450 (Table 3). Therefore, the number of registered cars explains 45% of the number of passengers variation. However, the Durbin–Watson value was 0.706, outside the range [1.0; 2.5]. Therefore, the errors have autocorrelation, and the residuals are not independent (and this is not desired).

**Table 3.** Model tests summary.

| Model | R | R Square | Adjusted R Square | Std. Error of the Estimate | Durbin-Watson |
|---|---|---|---|---|---|
| 1 | 0.671 | 0.450 | 0.445 | 1,330,450.985 | 0.706 |

Predictor: (Constant), *Xcar*; Dependent Variable: *Yt*.

The ANOVA test (analysis of variance) assesses whether there is a statistically significant difference between the regression model and the alternative model (the simple average of the values of the independent variable, hereinafter referred to as the alternative model) was included. In general, test a null hypothesis (H0) that indicates that the regression model is the same as the alternative model, against the alternative hypothesis (H1) that indicates that the models are different. If the null hypothesis is rejected, it is assumed that the regression model differs from the alternative model, and therefore, it can explain the analyzed phenomenon. The ANOVA data indicate that the model's F test is statistically significant (F = 96,561 and $p = 0.000 < 0.005$—in bold) (Table 4), that is, the model with the predictor by the number of registered cars can predict passenger demand.

**Table 4.** ANOVA results.

| | Model | Sum of Squares | df | Mean Square | F | Sig. |
|---|---|---|---|---|---|---|
| 1 | Regression | 170,922,833,351,082.750 | 1 | 170,922,833,351,082.750 | 96.561 | **0.000** |
| | Residual | 208,871,779,214,347.060 | 118 | 1770,099,823,850.399 | | |
| | Total | 379,794,612,565,429.800 | 119 | | | |

Dependent Variable: *Yt*; Predictor: (Constant), *Xcar*.

It is possible to observe that the coefficients are significant ($p = 0.000 < 0.005$—in bold) (Table 5), both for the intercept and the angular coefficient of the independent variable.

**Table 5.** Coefficients.

| Model | | Unstandardized Coefficients | | Standardized Coefficients | t | Sig. |
|---|---|---|---|---|---|---|
| | | B | Std. Error | Beta | | |
| 1 | (Constant) | 38,775,035.512 | 2,399,683.174 | | 16.158 | **0.000** |
| | Xcar | −25.205 | 2.565 | −0.671 | −9.827 | **0.000** |

Dependent Variable: *Yt*.

Therefore, the obtained model is $Y^t = 38,775,035.12 - 25.205X^t_{car}$. The angular coefficient shows that the number of passengers reduces by approximately 25.02 with each new car registered. This finding confirms the hypothesis of negative impact on the variable. The absence of outliers was verified and can occur when the minimum and maximum range for standardized residuals must be between $[-3; +3]$. There are no outliers observed in the model's analyzed numbers (Table 6).

**Table 6.** Residuals Statistics.

| | Minimum | Maximum | Mean | Std. Deviation | N |
|---|---|---|---|---|---|
| Predicted Value | 14,020,954.000 | 18,009,194.000 | 15,224,644.950 | 1,198,468.326 | 120 |
| Residual | −3,030,565.750 | 2,703,121.750 | 0.000 | 1,324,849.061 | 120 |
| Std. Predicted Value | **−1.004** | **2.323** | 0.000 | 1.000 | 120 |
| Std. Residual | **−2.278** | **2.032** | 0.000 | 0.996 | 120 |

Dependent Variable: *Yt*.

Data normality was checked graphically by the histogram (Figure 2) and the p-p plot (Figure 3). The independent variable frequency analysis must be close to the normal curve in the histogram, which occurs in fact.

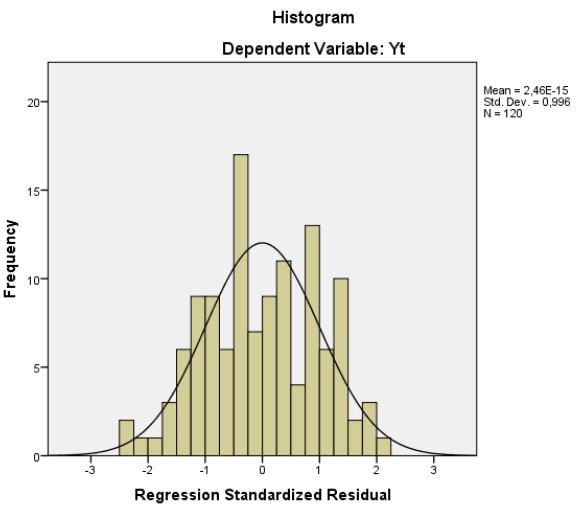

**Figure 2.** Histogram.

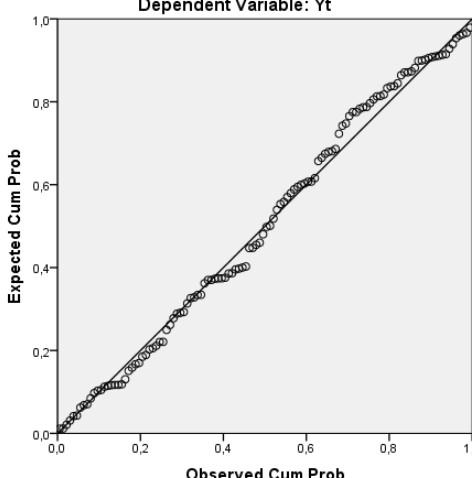

**Figure 3.** P-P Plot graphic.

The P-P Plot can verify the residual's normality when the residual's cumulative probability equals the observed cumulative probability, which represents the bisector line of the even quadrants. There is an overlap observed, and the residuals are normal.

The last prerequisite of the regression model is that the error variance is constant. If this occurs, the model is said to be homoscedastic [40]. Graphically, the error dispersion distribution must be similar to a rectangle. If a conic figure appears, then the model is heteroscedastic. The graph (Figure 4) shows that the homoscedasticity requirement is not guaranteed, as the figure is distinct from a rectangle.

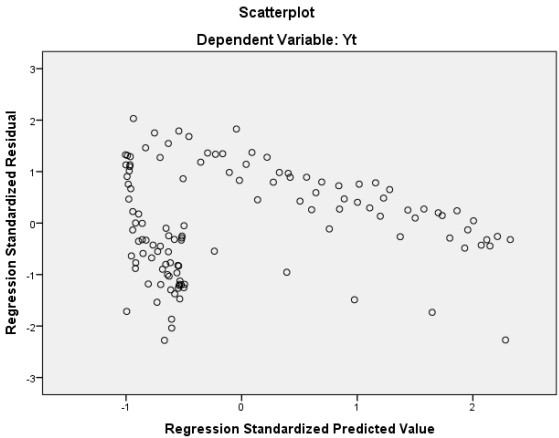

**Figure 4.** Homoscedasticity graph.

Simple linear regression showed that the number of registered cars predicts the demand for public transport [$F(1.118) = 95,651$, $p < 0.001$; $R2 = 0.450$]. The forecast model corresponds to $Y^t = 38,775,035.12 - 25.205 \times X_{car}^t$. The Shapiro–Wilks test was performed to verify the normality of the data and the results are $W = 0.98402$ and $p$-value $= 0.1757$ (>0.05, the null hypothesis that the data are normal cannot be rejected).

### 3.2. Relationship between Number of Registered Motorcycles and System Demand

Analysis model:

$$M2: \ Y^t = \beta_0 + \beta_1 X_{mot}^t \tag{3}$$

There is required $n \geq 20$ for each predictor variable. A single predictor variable and 120 (|T| = 120) observations satisfy this requirement, consequently. The linear relationship between the independent ($X_{mot}^t$) and dependent ($Y^t$) variables is observable in Figure 5.

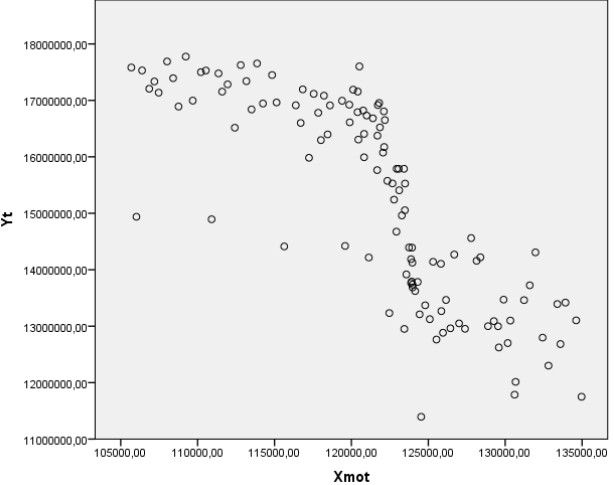

**Figure 5.** Linear relationship between independent variable ($X_{mot}^t$) and dependent variable ($Y^t$).

It is possible to observe that the relationship is not perfectly linear between the variables, although there is a visual relationship. Table 7 illustrates the correlation test between the variables.

**Table 7.** Correlation analysis results.

| | | *Yt* | *Xmot* |
|---|---|---|---|
| Pearson Correlation | *Yt* | 1.000 | **−0.798** |
| | *Xmot* | **−0.798** | 1.000 |
| Sig. (1-tailed) | *Yt* | | **0.000** |
| | *Xmot* | **0.000** | |
| N | *Yt* | 120 | 120 |
| | *Xmot* | 120 | 120 |

The correlation test between the variables shows that the Pearson's coefficient is strong (−0.798) and negative indeed, which means, as the number of registered motorcycles increases, the number of passengers decreases. The correlation *e* is statistically significant ($p = 0.000 < 0.005$). Regarding independent residues, a model's prerequisites are don't have autocorrelation between the residuals (difference between the observed value and the value predicted by the model). In this case, the Durbin–Watson test must have a value between 1.0 and 2.5. Table 8 summarizes these test results.

**Table 8.** Model Summary.

| Model | R | R Square | Adjusted R Square | Std. Error of the Estimate | Durbin-Watson |
|---|---|---|---|---|---|
| 1 | 0.798 | 0.637 | 0.634 | 1,081,335.229 | 1.059 |

Predictor: (Constant), *Xmot*; Dependent Variable: *Yt*.

The analyzed data show that the R-squared is 0.637. Therefore, the number of registered motorcycles explains 63.7% of the number of passengers variation. The value of the Durbin–Watson test was 1.059, still within the range [1.0; 2.5], which precludes autocorrelation in errors. Data from the ANOVA show that the model's F test is statistically significant (F = 206,809 and $p = 0.000 < 0.005$—in bold) (Table 9), which means that the model with the number of registered motorcycles as a predictor can forecast the passenger demand.

**Table 9.** ANOVA results.

| Model | | Sum of Squares | Df | Mean Square | F | Sig. |
|---|---|---|---|---|---|---|
| 1 | Regression | 241,818,878,815,977.380 | 1 | 241,818,878,815,977.380 | 206,809 | **0.000** |
| | Residual | 137,975,733,749,452.440 | 118 | 1,169,285,879,232.648 | | |
| | Total | 379,794,612,565,429.800 | 119 | | | |

Dependent Variable: *Yt*; Predictor: (Constant), *Xmot*.

It was possible to observe that the coefficients are significant ($p = 0.000 < 0.005$—in bold) (Table 10), for the intercept and the slope of the independent variable.

**Table 10.** Coefficients.

| Model | | Unstandardized Coefficients | | Standardized Coefficients | t | Sig. |
|---|---|---|---|---|---|---|
| | | B | Std. Error | Beta | | |
| 1 | (Constant) | 39,639,340.583 | 1,700,589.277 | | 23.309 | **0.000** |
| | *Xmot* | −201.179 | 13.989 | −0.798 | −14.381 | **0.000** |

Dependent Variable: *Yt*.

The model obtained is, therefore, $Y^t = 39,639,340.6 - 201.179 \times X^t_{mot}$. The angular coefficient shows that the number of passengers reduces by approximately 201.18 with each new motorcycle registered. This finding confirms the hypothesis of negative impact on the variable. The absence of outliers was verified and can be confirmed with the minimum and maximum range for standardized residues between $[-3; +3]$. There are outliers in the model's residuals observed (Table 11).

**Table 11.** Residuals Statistics.

|  | Minimum | Maximum | Mean | Std. Deviation | N |
|---|---|---|---|---|---|
| Predicted Value | 12,487,234.000 | 18,377,350.000 | 15,224,644.950 | 1,425,514.439 | 120 |
| Residual | −3,372,277.750 | 2,208,325.750 | 0.000 | 1,076,782.219 | 120 |
| Std. Predicted Value | **−1.920** | **2.212** | 0.000 | 1.000 | 120 |
| Std. Residual | **−3.119** | **2.042** | 0.000 | 0.996 | 120 |

Dependent Variable: *Yt*.

Data normality was checked graphically by the histogram (Figure 6) and the p-p plot (Figure 7). The independent variable frequency analysis must be close to the normal curve in the histogram, which occurs in fact.

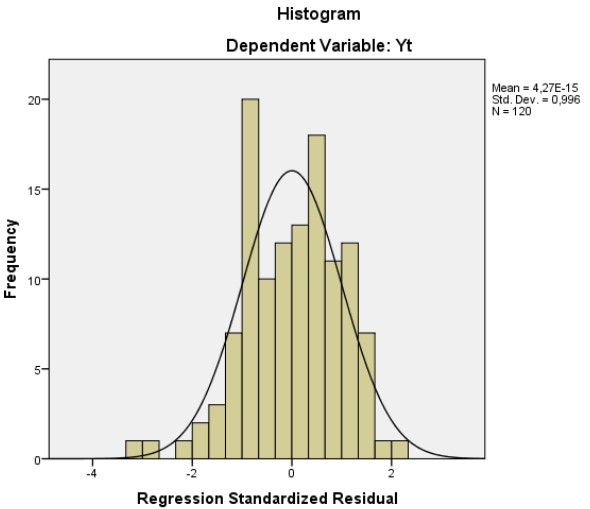

**Figure 6.** Histogram.

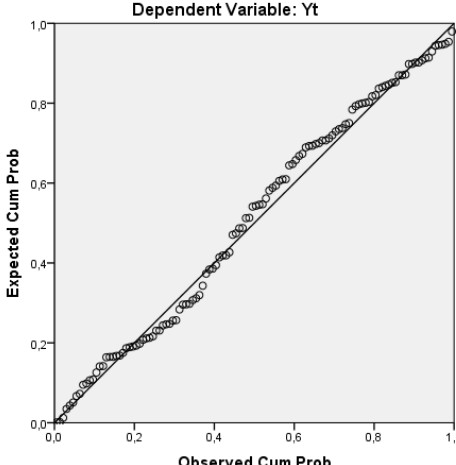

**Figure 7.** P-P Plot graphic.

The p-p plot can verify the residual's normality when the residual's cumulative probability equals the observed cumulative probability, which represents the bisector line of the even quadrants. There is an overlap observed even if it is not perfect, and the residuals are normal.

The last prerequisite of the regression model is that the error variance is constant. If this occurs, the model is said to be homoscedastic. Graphically, the error dispersion distribution must be similar to a rectangle. If a conic figure appears, then the model is heteroscedastic. The graph (Figure 8) shows that the homoscedasticity requirement is not guaranteed, as the figure is distinct from a rectangle.

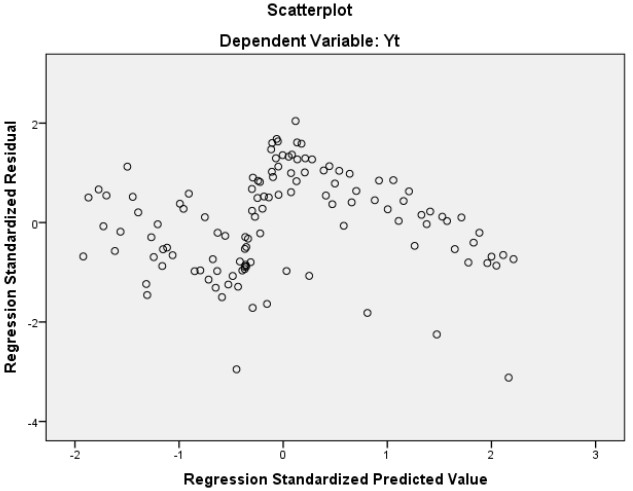

**Figure 8.** Homoscedasticity graph.

Simple linear regression showed that the number of registered motorcycles predicts the demand for public transport [$F_{(1.118)} = 206,809$, $p < 0.001$; $R^2 = 0.637$]. The forecast model corresponds to $Y^t = 396,393.40 - 201.18 \times X^t_{mot}$. The Shapiro–Wilks test was performed to verify the normality of the data and the results are W = 0.9805, *p*-value = 0.08425 (>0.05, the null hypothesis that the data are normal cannot be rejected).

### 4. Case of Study

The City of Curitiba (PR) started to be known worldwide in 1974 for BRT line implementation, locally called the integrated transport network (RIT), which is considered the ideal solution for transportation in accelerated growing cities. This transport system requires lower investments and less building time when compared to other options such as the subway, for example, since it uses buses circulating on existing roads with exclusive lanes and with fare payments made outside the vehicle before boarding. It becomes an attractive solution because it is faster, cheaper, and more flexible than a rail solution with practically the same level of service, and it offers more comfort and faster travel for public transport users.

BRT continues to be the main form of transportation in the city and, even with the population continuing to grow, it has been losing passengers year after year to other modes due to several factors such as climate conditions, fare, and fuel prices, and the ethical and aesthetical preferences of users who replace public transport by automobiles and motorcycles.

In addition to the seasonal effects, which reduce demand in December months, it is possible to observe a structural downward trend in passengers for the public transport service, lessening more than 30% in the aggregate demand for these lines in the last 10 years. It seeks, therefore, to identify which factors account for this dynamic, especially which forms of urban mobility are the protagonists in the capture of passengers from the public transport system.

The collected data record the aggregate number of passengers that used the BRT 203, 303, 502, 503, and 602 express lines between January 2010 and December 2019, totaling 120 periods. These lines together represent approximately 30% of the total passengers of Curitiba's transport system. Numbers were collected monthly, and there were no gaps noted.

## 5. Research Results and Discussion

Research results reveal a natural substitution effect between cars and motorcycles and the demand for transport. As described in Section 3, the public transport substitution by cars (Model 1) and motorcycles (Model 2) is under the effect of fuel prices. The result confirmed the hypothesis that, with more expensive fuel, the user leaves mostly the car (or the motorcycle in a much lower proportion) at home and goes to public transport. Each increase of R$ 0.10 (10 cents) in fuel price makes the number of monthly passengers grow by around 5.000 in the six studied BRT lines. It is important to note that this substitution is not definitive since passengers still have a car (or a motorcycle) and can use it back if the fuel price goes down.

This observation states one important dichotomy to sustainable urban management because when fuel price increases also affect the fare price since BRT in Curitiba uses diesel. The municipality occasionally has to decide between keeping the fare price frozen artificially (and subsiding the difference from the budget) or raising it accordingly. Depending on this decision, there will be more passengers on buses or more drivers in cars.

In the period analyzed by the present research, it is possible to infer that the city administration sought to keep ticket prices as stable as possible for political, social, and economic reasons. Politically because of the large numbers of voters who depend on public transport, socially because passengers are low-income in general and economically to maintain the financial balance of the transport system. This balance is sensitive and tenuous, exacerbated by budget constraints that prevent significant investments, whether in the public transport system or on road infrastructure.

In addition, and confirming the findings described above, the statistical analysis demonstrated the effect of the number of cars and motorcycles on the public transport demand. Both models described in Section 3 (Models 1 and 2) passed and had statistical significance. In Model 1, it is interesting to observe that each additional car in circulation removes only 25 passengers (or BRT travels) from the system. In Model 2, each new motorcycle in circulation takes out 201 passengers (or BRT travels) from the system.

This difference comes from the fact that the motorcycle is better for many trips on short routes, which would be expensive in public transport since each trip would cost the price of a ticket. Given this, it is possible to affirm that the replacement by motorcycles tends to be definitive, especially in large urban centers where, in addition to the cost, mobility is complicated, and parking areas are few and expensive. In conclusion, workers involved with all kinds of delivery and last-mile services are moving off from the system as soon as they can buy a motorcycle or a bicycle.

Cars, in turn, are used for long distances, usually by passengers who use all if not the entire route of the BRT line. These are passengers who take only one round trip a day to work or school and whose distances are long, making the ticket price attractive in terms of the cost of getting around by other means of their own and the time loss in traffic, the risks of accidents and robberies, and some further expenses associated with leaving the car parked all day on the street or in paid places.

In a previous article, the authors of [41] observed that the number of boarded passengers at Curitiba's BRT system is almost stable, with significant variation only during school periods. Therefore, it is possible to conclude that students are the most regular passengers in public transportation. They do not use or take into consideration other types of transport. One possible explanation resides on partial or total gratuity granted for students turning other options not economically attractive. Fare price still being a very, or the most,

significant factor for passengers during their decision-making about transportation for all different groups (industry, commerce and service workers, and students).

These conclusions can be explored positively by the city in terms of public policy if they want to equalize the public system demand and investments in urban infrastructure to attend to all population needs equally. Consequently, since the municipality can monitor all this information and results shown here, this data must be handled and managed intelligently to support proper transport policies and other urban development investments more democratically and sustainably.

## 6. Conclusions

Large cities' administrations are facing a series of complex variables aggravated by the scarcity of available resources to invest assertively in the adequate transport infrastructure that meets, in the most equitable way, the citizens' transport needs for the most diverse income categories. Unfortunately, those who need transportation more are the ones who have the least options. They live in areas too far away to walk or cycle, the ticket price in terms of comfort, safety, and reliability is costly, and the use of own vehicles becomes impractical not only because of acquisition and maintenance costs but also because of fuel price and the time loss in traffic jams at peak hours.

The number of private vehicles already borders the physical limit of available space and is causing, in addition, externalities that involve air and noise pollution, and there is no way for the municipality to expand the existing streets. The city administration may believe that bicycles would replace cars and motorcycles through incentives in taxation and investments in exclusive bike lanes all around the city. However, fewer passengers raise the fare price and, consequently, an increase in subsidies or gratuity that still consumes money from the municipal budget. A minimum level of paying passengers is required for public transport to be minimally viable for both the municipality and the companies operating the system.

The city of Curitiba has an automatic ticketing system that covers 67.67% of the total number of users, and the remaining paid tickets are controlled electronically, in real-time, at the boarding stations. Buses are all electronically monitored, and the company that manages and invigilates public transportation in the city (URBS) supervises the occupation of the cars. This company also defines the ticket price. The other data such as climate, vehicles in circulation, new bicycles, and the fuel price are also available electronically. They are updated practically daily and are simple to monitor as well. As mentioned in the literature review, other variables, and latent variables, like comfort, safety, service level, punctuality, must be included and learned from other cities' experiences [42].

Once all the necessary information for decision-making is available, the city may decide more assertively about which transport infrastructure to invest in, based on data intelligence tools. Planning is one of the main instruments for managing municipalities, city halls, and public organizations. Such concepts are related to decision-making resources and the application of activities aimed at managing acts [37]. The municipality can define transport policies according to the intended future for the city in terms of transport and urban form.

The research contributions are directed to the City of Curitiba, which can take advantage of the results obtained, thus facilitating the city managers' decisions and the citizens who use public transport and mobility options. For researchers on topics related to public transportation, the research allowed yet another advance in science based on the studies of its variables. The research limitation portrays the non-generalization of its results to other cities and, on the other hand, allows the application of these methods in other cities, for example, Bogota, which uses Curitiba's BRT system on a large scale [7]. Despite criticisms and proposals for improvement, the BRT remains a worldwide success. According to [43], more than 180 cities on all continents have already adopted the system. More than 33 million passengers use the BRT per day at all these locations.

For these reasons, the smart city and strategic digital city projects (SDC) [19,20,38,44,45] concept application seems adequate not only to monitor important and sensitive variables that affect the users' choice of a particular modal but mainly to integrate buses, vehicles, and bicycles in a balanced model able not only to offer transport solutions but also to attract more people to public transport and other clean and sustainable transport means. The SDC can also lead to appropriate investment decisions by the municipality according to systemic planning that can offer options for transport with quality, comfort, safety, fluidity, and price in an equitable way for all citizens. The SDC project involves four subprojects [19,20,38]: strategy, information, services, and technology. Curitiba has an existing and implemented digital strategy regarding mobility, and the variables (information subproject) explored in this article are available for decision-makers and citizens. The present article helps to clarify that there are relevant relations between them that must be incorporated by the city administration and must be available to all passengers to assist in making the best decision about transport options. Citizens can also follow digitally if the city investments are consonant with the best interests.

**Author Contributions:** Conceptualization, L.A.W.F., D.A.R. and T.A.G.; methodology, L.A.W.F., D.A.R. and T.A.G.; software, T.A.G.; validation, L.A.W.F., D.A.R. and T.A.G.; formal analysis, L.A.W.F., D.A.R. and T.A.G.; investigation, L.A.W.F.; resources, L.A.W.F.; data curation, L.A.W.F.; writing—original draft preparation, L.A.W.F.; writing—L.A.W.F.; visualization, L.A.W.F.; supervision, D.A.R.; project administration, L.A.W.F.; funding acquisition, not applicable. All authors have read and agreed to the published version of the manuscript.

**Funding:** This research received no external funding.

**Institutional Review Board Statement:** Not applicable.

**Informed Consent Statement:** Not applicable.

**Data Availability Statement:** Not applicable.

**Acknowledgments:** CNPq, PUCPR, FAE.

**Conflicts of Interest:** The authors declare no conflict of interest.

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
