# Peer review of "Data Intelligence in Public Transportation: Sustainable and Equitable Solutions to Urban Modals in Strategic Digital City Subproject"

_sustainability, doi:10.3390/su14084683_

Round 1

Reviewer 1 Report

The study used the aggregate number of passengers that used the main Curitiba’s (Brazil) BRT lines to find the relationship between the number of passengers and the number of private vehicles, public transport fare, and fuel price. It is a good paper but should address the following concerns.

I suggest the authors clarify the two statistical regression models stated in the abstract. What type of regression?

I suggest the authors check other models, as I doubt that a simple linear regression is the best model. See, for example, Figure 1.

There are several statistical tests to check the normality of a distribution. I suggest the authors use at least one of them.

The authors should mention that some other variables/ latent variables might influence the ridership. See, for example, doi.org/10.1016/j.tra.2020.10.020 as they discussed convenience, comfort, reliability, and security.

The authors highlight a conclusion; urban management and strategic digital city projects can be more balanced and assertive in transport infrastructure investments and citizen services provision. I expect to see a more robust discussion in the paper to support this conclusion. 

The data was collected before the COVID-19 pandemic. However, this pandemic had / can significantly impact public transportation. See, for example, doi.org/10.1016/j.erss.2020.101666 as they discussed some opportunity windows.

Following the previous comment, in the discussion part, the authors stated that the number of boarded passengers at Curitiba's BRT system is almost stable (except for school periods) even during the COVID-19 pandemic. Since the discussion is about policy-making, it is necessary to expand this issue further, given the findings of other countries.

As a minor comment, I suggest authors write the exact names of the dependent/independent variables on the axes in the scatter plots (figures).

Also, some moderate English changes/corrections required.

Author Response

Point 1: I suggest the authors clarify the two statistical regression models stated in the abstract. What type of regression?

Response 1: The expression “two statistical regression models” has been replaced with “two linear regression models” in lines 19 and 73.

Point 2: I suggest the authors check other models, as I doubt that a simple linear regression is the best model. See, for example, Figure 1.

Response 2: As suggested, an additional verification was carried out for model 1, whose reporting is limited to this document.

Two variations for model 1 are reported below. Depending on the distribution of the data, an alternative polynomial model of degree 2 was evaluated.

Alternative model (degree 2 polynomial) 1:

Model

R

R Square

Adjusted R Square

Std. The error of the Estimate

Durbin-Watson

M1’

,709a

,504

,4961

1279000

,6396

a. Predictor: (Constant),

b. Dependent Variable:

Comparison between M1 e M1’

  • H0: M1=M1’
  • H1: M1≠M1’

Model

Df

RSS

Df

SSQ

F

Pr(>F)

M1

119

1.93E+14

M1'

118

1.88E+14

1

5.11E+12

3.1275

0.07963

As Pf(>F)=0.07963 < 0.05, the null hypothesis that M1=M1' cannot be rejected. Therefore, a linear relationship was adopted. Additionally, the interpretation of the linear model is more suitable for the research context.

Point 3: There are several statistical tests to check the normality of a distribution. I suggest the authors use at least one of them.

Response 3: The Shapiro Wilks test was included to verify the normality of the data. The test result is shown in lines 286 (for the M1 model) and 354 (for the M2 model)
In line 286, it was inserted: “W = 0.98402 and p-value = 0.1757 (>0.05, the null hypothesis that the data are normal cannot be rejected)”

In line 354, it was inserted: “W = 0.9805, p-value = 0.08425 (>0.05, the null hypothesis that the data are normal cannot be rejected)”.

Point 4: The authors should mention that some other variables/ latent variables might influence the ridership. See, for example, doi.org/10.1016/j.tra.2020.10.020 as they discussed convenience, comfort, reliability, and security.

Response 4: Mention added on line 462.

Point 5: The authors highlight a conclusion; urban management and strategic digital city projects can be more balanced and assertive in transport infrastructure investments and citizen services provision. I expect to see a more robust discussion in the paper to support this conclusion.

Response 5: Paragraph added on line 490.

Point 6: The data was collected before the COVID-19 pandemic. However, this pandemic had / can significantly impact public transportation. See, for example, doi.org/10.1016/j.erss.2020.101666 as they discussed some opportunity windows.

Response 6: The previous study from the authors was done regarding this particular issue and the reference and comment are at line 423. https://doi.org/10.1016/j.jum.2021.04.002

Point 7: Following the previous comment, in the discussion part, the authors stated that the number of boarded passengers at Curitiba's BRT system is almost stable (except for school periods) even during the COVID-19 pandemic. Since the discussion is about policy-making, it is necessary to expand this issue further, given the findings of other countries.

Response 7: Same as per Point 6. The previous study from the authors was done regarding this particular issue and the reference and comment are at line 423. https://doi.org/10.1016/j.jum.2021.04.002

Point 8: As a minor comment, I suggest authors write the exact names of the dependent/independent variables on the axes in the scatter plots (figures).

Response 8: Figures are created from SPSS and it is not possible to modify. In Figure 1 and Figure 5 the independent variable is on the horizontal axis, and the dependent variable is on the vertical axis.

Reviewer 2 Report

Data intelligence in public transportation:  managing and investing for sustainable and equitable solutions to urban modals in strategic digital city subproject

General comments

the introduction and review of the literature should further highlight the "research gap" finally describing the objectives pursued in the research.

The methodology followed is also not well defined by the authors. It should be better explained what type of analysis is going to be carried out and what type of variables and coefficients are going to be calculated to check the goodness of the correlations.

Results seem to me to be well discussed in section 5 and the conclusions of the investigation are consistent with the results obtained.

Specific comments

Titles should be in lowercase, with the first letter of each word capitalized.

Line 68. “BRT lines” should be defined the first time is cited in the abstract and in the main manuscript. (Is it Bus rapid transit?)

Line 150. NCD should be defined the first time is cited. (are they non-communicable disease?)

Lines 199-205. Check the content, it seems that “data collected between January 2010 and December 2019 monthly”,  is duplicated.

Line 228. Consider merging sections 3 and 4 into one section (methodology)

Line 244. “Pearson's coefficient is moderate to strong”. Add reference to justify it.

Line 250. “a model's prerequisites are don't have autocorrelation between the residuals”. Check the grammar, it does not sound good.

Table 3. Check the decimal separation. It seems that you have mixed points with the commas. The same happens in table 2, 4 and 5 (please check the whole manuscript).

In table 3 “model test summary”, R is 0.671 and a letter a is added. What does it mean?. It is not clear. It seems it has nothing to do with a. Predictor or b. Dependent variable.

Line 259 “ANOVA data”. Could you explain a little bit what is this for (analysis of variance), or add a reference ?

Table 4. ANOVA a. The meaning of the “a” is again difficult to understand. Does it have something to do with “a. Dependent variable or b.Predictor”?

Line 291. “the model is said to be homoscedastic”. Could you add a reference?

Table 11 and others. Results with different number of decimal places are used. Consider always using the same number of decimals. E.g. 3.

References should follow sustainability template.  

  1. Author 1, A.B.; Author 2, C.D. Title of the article. Abbreviated Journal Name Year, Volume, page range.

And so on.

Author Response

Point 1: The methodology followed is also not well defined by the authors. It should be better explained what type of analysis is going to be carried out and what type of variables and coefficients are going to be calculated to check the goodness of the correlations.

Response 1: A better explanation of the type of analysis was inserted in the methodology section (line 182), as well as the type of variables to quantify the causality for the effect of substituting demand for public transport.

Point 2: Titles should be in lowercase, with the first letter of each word capitalized.

Response 2: Done.

Point 3: “BRT lines” should be defined the first time is cited in the abstract and the main manuscript. (Is it Bus rapid transit?).

Response 3: Yes. Included.

Point 4: NCD should be defined the first time is cited. (are they non-communicable diseases?

Response 4: Yes. Included.

Point 5: Check the content, it seems that “data collected between January 2010 and December 2019 monthly”,  is duplicated.

Response 5: Yes. Excluded.

Point 6. Consider merging sections 3 and 4 into one section (methodology).

Response 6: Sections are now inverted due to other reviewer indications. Section 3 now is Methodology and Section 4 presents Curitiba as the Case of Study.

Point 7: Pearson's coefficient is moderate to strong”. Add a reference to justify it.

Response 7: Done.

Point 8: “a model's prerequisites are don't have autocorrelation between the residuals”. Check the grammar, it does not sound good.

Response 8: Done.

Point 9: Check the decimal separation. It seems that you have mixed points with the commas. The same happens in Tables 2, 4, and 5 (please check the whole manuscript).

Response 9: Done.

Point 10: In table 3 “model test summary”, R is 0.671, and a letter a is added. What does it mean?. It is not clear. It seems it has nothing to do with a. Predictor or b. Dependent variable.

Response 10: It was just a typo, which has already been corrected. The parameter R refers to the degree of explanation of the model, that is, the extent to which the dependent variable can explain the variance of the dependent variable. There is no relationship with the parameters of the model used.

Point 11: “ANOVA data”. Could you explain a little bit what is this for (analysis of variance), or add a reference?

Response 11: The explanatory text was included (line 242): “the ANOVA test (analysis of variance) assesses whether there is a statistically significant difference between the regression model and the alternative model (the simple average of the values of the independent variable, hereinafter referred to as the alternative model) was included. In general, test a null hypothesis (H0) that indicates that the regression model is the same as the alternative model, against the alternative hypothesis (H1) that indicates that the models are different. If the null hypothesis is rejected, it is assumed that the regression model differs from the alternative model, and therefore, it can explain the analyzed phenomenon”.

Point 12: Table 4. ANOVA a. The meaning of the “a” is again difficult to understand. Does it have something to do with “a. Dependent variable or b.Predictor”?

Response 12: Again, it was just a typo, which has already been corrected. The parameter R refers to the degree of explanation of the model, that is, the extent to which the dependent variable can explain the variance of the dependent variable. There is no relationship with the parameters of the model used.

Point 13: “the model is said to be homoscedastic”. Could you add a reference?

Response 13: Reference included as suggested.

Point 14: Table 11 and others. Results with a different number of decimal places are used. Consider always using the same number of decimals. E.g. 3.

Response 14: All are correct as suggested.

Point 15: References should follow Sustainability template.

Response 15: All are correct as indicated.

Reviewer 3 Report

sustainability-1631222 is quite a good paper, that needs only some modifications in order to be publishable:

  1. Change the title, it is too long. It looks more like an abstract in the current form;
  2. I feel like the introduction may include some additional arguments to make the points of the authors about the sustainability in urban premises stronger:

(21st Century solutions for urban mobility):

Smolnicki, P.M., Sołtys, J. (2016). Driverless Mobility: The Impact on Metropolitan Spatial Structures. Procedia Engineering 161, pp. 2184-2190

Ströhle, P., et al. (2019). Leveraging customer flexibility for car-sharing fleet optimization. Transportation Science 53(1), pp. 42-61

Bruzzone, F., et al. (2021). The combination of e-bike-sharing and demand-responsive transport systems in rural areas: A case study of Velenje. Research in Transportation Business & Management 40, 100570

(Urban logistics sustainable management):

Goldman, T., Gorham, R. (2006). Sustainable urban transport: Four innovative directions. Technology in Society 28(1-2), pp. 261-273

Quak, H.J., De Koster, M.B.M. (2009). Delivering goods in urban areas: How to deal with urban policy restrictions and the environment. Transportation Science 43(2), pp. 211-227

Nocera, S., Cavallaro, F. (2017): A Two-Step Method to Evaluate the Well-To-Wheel Carbon Efficiency of Urban Consolidation Centres. Research in Transportation Economics 65, pp. 44-55

(Urban tourism management):

McKercher, B. (1993). The unrecognized threat to tourism. Can tourism survive 'sustainability'? Tourism Management 14(2), pp. 131-136

Paskaleva-Shapira, K.A. (2007). New paradigms in city tourism management: Redefining destination promotion. Journal of Travel Research 46(1), pp. 108-114

Cavallaro, F., et al. (2017): Policy Strategies for the Mitigation of GHG Emissions caused by the Mass-Tourism Mobility in Coastal Areas. Transportation Research Procedia  27: 317-327

3. I would consider the inversion of sections 3 and 4. I feel like the method should be explained first, and then the city of Curitiba offered as a case study. Please adjust the text accordingly;

4. Finally, the discussion should include the generalization margin of the model outside of Curitiba. Please add a brief paragraph about this.

Author Response

Point 1: Change the title, it is too long. It looks more like an abstract in the current form.

 Response 1: It's short as possible now, without losing the intended meaning from the author's perspective. The title now is: Data Intelligence in Public Transportation: Sustainable and Equitable Solutions to Urban Modals in Strategic Digital City Subproject.

Point 2: I feel like the introduction may include some additional arguments to make the points of the authors about the sustainability in urban premises stronger.

Response 2: Reviewer 2 brought great suggestions that now are contributions to the article. Six (of nine) suggested articles are included in the Introduction (lines 39, 42, 62 and 66).

Point 3: 3. I would consider the inversion of sections 3 and 4. I feel like the method should be explained first, and then the city of Curitiba offered as a case study. Please adjust the text accordingly.

Response 3: It's done as suggested. Section 4 now is section 3 and the former sections 4.1 and 4.2 now are 3.1 and 3.2. The new Section 4 explains the case of Curitiba and the details about collected data for the study.

Point 4: 3. Finally, the discussion should include the generalization margin of the model outside of Curitiba. Please add a brief paragraph about this.

Response 4: Included as requested (line 478).

Round 2

Reviewer 1 Report

The authors could address my comments properly.

Reviewer 2 Report

The authors have followed my recommendations one by one . Just comment that section 5 should be named "Research results and discussion"  due to the order in which both are obtained.

Reviewer 3 Report

Article adequately revised, looking good now. Ready for publication